# Preparation of Boron Nitride-Coated Carbon Fibers and Synergistic Improvement of Thermal Conductivity in Their Polypropylene-Matrix Composites

**DOI:** 10.3390/polym11122009

**Published:** 2019-12-04

**Authors:** Arash Badakhsh, Woong Han, Sang-Chul Jung, Kay-Hyeok An, Byung-Joo Kim

**Affiliations:** 1Research Laboratory for Multifunctional Carbon Materials, Korea Institute of Carbon Convergence Technology, Jeonju 54853, Korea; badakhsh.arash@gmail.com (A.B.); shareyi@kctech.re.kr (W.H.); 2Department of Mechanical Design Engineering, Chonbuk National University, Jeonju 54896, Korea; 3Department of Organic Materials and Fiber Engineering, Chonbuk National University, Jeonju 54896, Korea; 4Department of Environmental Engineering, Sunchon National University, Sunchon 57922, Korea; jsc@sunchon.ac.kr; 5Department of Carbon and Nano Materials Engineering, Jeonju University, Jeonju 55069, Korea

**Keywords:** carbon fiber, boron nitride, coating, polymer–matrix composites, thermal properties, electrical resistivity

## Abstract

The purpose of this study is to prepare boron nitride (BN)-coated carbon fibers (CF) and to investigate the properties of as-prepared fibers as well as the effect of coating on their respective polymer–matrix composites. A sequence of solution dipping and heat treatment was performed to blanket the CFs with a BN microlayer. The CFs were first dipped in a boric acid solution and then annealed in an ammonia–nitrogen mixed gas atmosphere for nitriding. The presence of BN on the CF surface was confirmed using FTIR, XPS, and SEM analyses. Polypropylene was reinforced with BN–CFs as the first filler and graphite flake as the secondary filler. The composite characterization indicates approximately 60% improvement in through-plane thermal conductivity and about 700% increase in the electrical resistivity of samples containing BN-CFs at 20 phr. An increase of two orders of magnitude in the electrical resistivity of BN–CF monofilaments was also observed.

## 1. Introduction

With the swift advances in the production of compact electronics, the thermal management of such units has become of high interest to engineers and designers [1,2,3]. Accordingly, the fabrication of thermally conductive and electrically insulating materials for this purpose has been targeted in several studies to efficiently enhance the heat rejection performance from the system while utilizing the advantages of novel materials, including lower cost and weight, as well as higher chemical and physical stability under harsh working conditions. Despite their low intrinsic thermal conductivity, polymers are good alternatives for this application, mainly because of their high resistance to corrosion and fouling, ease of processing, recyclability, low cost, and light weight [4,5,6]. To enhance the thermal conductivity in polymers, highly conductive fillers, such as carbon fibers (CFs), carbon nanotubes (CNTs), graphene nanosheets, and graphite powder, as well as ceramic and metallic particles can be used [7,8,9,10,11]. Among these, CF has drawn more attention in composite design where anisotropic high thermal conductivity, large mechanical load transfer, and light weight are desirable. However, the inertness of the CF surface and the difference in its surface energetics with polymers play a key role in the integrity and properties of the final multicomponent system [12,13,14,15]. Therefore, the challenge addressed here is to enhance the interfacial adhesion between the matrix and the filler using the proper interfaces to obstruct the boundary phonon scattering while buffering the electrical conductivity of CFs for a better and safer performance in electronic units. Owing to their high thermal conductivity, low electrical conductivity, and high-temperature stability, ceramics are regarded as promising materials where such properties are needed [16]. Decoration of CF with ceramic films and/or particles is one of the ways used to form a superior thermally conductive CF-based composite with an electrically buffering coating [17]. There are a variety of methods to fabricate ceramic coatings, including sintering [18,19], laser cladding [20], dip-coating [21,22,23], chemical vapor deposition (CVD) [24,25,26,27], magnetron sputtering [28,29], or a sequence of all or some of these processes. Although sintering and CVD are considered to be the primary methods for the large-scale fabrication of ceramic-coated fibers, they impose several restrictions, including high cost, complexity, and difficulty of operation, as well as enormous energy consumption. On the other hand, the dip-coating method has been known for its simplicity and safety.

In the present study, boron nitride (BN) was selected as the target ceramic coating, as it possesses desirable properties, including low density, high thermal conductivity (~320 W/mK) and electrical resistivity [30,31,32]. Lii et al. reported a similar process for BN deposition on CF and graphite substrate, composed of a boric acid–urea solution for boriding (or boronizing) and ammonia-assisted nitriding in a resistance furnace. They analyzed the effect of various compositions (via molarity control) of boriding solution and nitriding temperature (up to 1000 °C) on the morphology of ceramic coatings [21]. Following a similar dip-coating process of BN on CF substrates, Zhou et al. studied the effect of BN coatings on the dielectric properties of paraffin-matrix composites reinforced with treated CFs [23]. Here, we report the preparation and characterization of BN–CFs using only a gaseous source of nitrogen in a high-temperature (1400 °C) furnace. Moreover, to the best of our knowledge, there have not been any reports on the improvement of thermal conductivity of polymer–matrix composites (PMCs) reinforced with BN–CFs. Polypropylene (PP) was used as the matrix mainly due to its wide commercial availability and good recyclability [33,34]. In this study, to further contribute to the formation of conductive pathways within the matrix, graphite flakes (GFs) were employed as a cost-effective secondary filler, which can lower the percolation threshold of the fillers due to their two-dimensional geometry. The main objectives of the present study are as follows:To simplify the synthesis of a BN microlayer on CFs by removing the adverse environmental effects of using urea in the boriding process, hence carrying out nitriding by using only a gaseous precursor in a high-temperature furnace.To measure the effects of BN coating on the thermal conductivity and electrical resistivity of the final PMC.To propose BN–CFs incorporated with GF as the reinforcing filler system for the fabrication of functional PMCs.

## 2. Experimental Details

### 2.1. Materials

The carbon fibers studied in this work, supplied by Toray Co., Tokyo, Japan, have a thermal conductivity of about 10.46 W/mK and were polyacrylonitrile (PAN)-based (T-300, 3 K). The fibers were chopped into one-inch threads before the treatment. Boric acid (H_3_BO_3_, 99.5%–pure, MW = 61.83 g/mol) and methanol were the reagent grades, purchased from Deajung Chemicals Co., Namyangju-si, Korea. The natural graphite powder (flakes, 99% carbon basis, +50 mesh particle size ≥ 80%) was provided by Sigma-Aldrich Chemicals Co., St. Louis, MO, USA. The matrix was polypropylene (SJ 150), supplied by Lotte Chemical Corp., Seoul, Korea. With the exception of the CFs, all the materials were used as received and without further purification.

### 2.2. Preparation of BN-Modified CFs

Firstly, the carbon fibers were immersed in acetone for 24 h, cleaned with distilled water several times, and then dried in an oven at 80 °C for 12 h. The obtained CFs were then dipped in the precursor solution with boric acid dissolved in methanol in a 1:3 mass ratio. After being ultrasonically agitated for 30 min, the treated carbon fibers were carefully parted from the solution and put into an oven at 80 °C, until completely dried. Afterwards, the borided CFs were put into a zirconia crucible and mounted inside the heating tube. The furnace was heated to 1400 °C under flowing nitrogen (N_2_) at 200 cc/min. The chamber was then maintained at 1400 °C for 60 min, while ammonia (NH_3_) gas (1000 ppm) was introduced at 100 cc/min. Ammonia acted as promoter and an extra source of nitrogen to obtain better solubility in carbon fiber substrate [35,36]. After 1 h, the ammonia gas flow was stopped and the samples were cooled down under N_2_ atmosphere. The heating and cooling rates were 10 °C/min. The preparation process of BN–CFs is illustrated in Figure 1.

### 2.3. Fabrication of PMC Samples

First, 120 g of polypropylene pellets were poured into the mixing chamber of an internal mixer at a temperature of 180 °C. The rotation of the mixing shafts was kept at 20 rpm. After the polymer was completely melted, the BN–CFs (at 20 phr) were added and mixed for 30 min. To facilitate the formation of conductive pathways and to reduce the phonon scatterings, GF (at 20 phr) was employed as the secondary (contribution) filler. The pasty composite was then extracted and compression-molded into 20 × 20 × 1 mm square samples. The molding pressure was controlled by a manual compressor and increased gradually to allow the remaining air in the composite to leave. During the molding process, the pressure was kept at 200 bar while the temperature was reduced by natural convection from 190 to 100 °C. The samples were then extracted and cooled down at room temperature.

### 2.4. Characterizations

#### 2.4.1. Surface Characteristics

Fourier-transform infrared spectroscopy (FTIR, Nicolet iS10, Thermo Fisher Scientific Inc., Waltham, MA, USA) was carried out to determine the functional groups present on the surface of the treated CFs. FTIR samples were prepared via potassium bromide (KBr) pelleting, and the analysis was performed at room temperature (298 ± 1 K) under air atmosphere and for 4000–1000 cm^−1^ wavelength range. Various samples were prepared and analyzed to ensure the consistency of the acquired FTIR spectra.

X-ray photoelectron spectroscopy (XPS, PHI 5000 VersaProbe II, ULVAC-PHI, Inc., Chigasaki, Japan) was used to further identify the surface chemistry of the BN-modified CFs. Unless otherwise specified, the X-ray anode was run at over 5 W, and the high voltage was kept at 5.0 kV. The energy resolution was fixed at 0.50 eV to ensure sufficient sensitivity. The base pressure of the analyzer chamber was about 5 × 10^−8^ Pa. Both the whole spectra (0–1200 eV) and narrow ones for all the elements were recorded with a very high resolution. Binding energies were calibrated with containment carbon (C_1s_ = 284.6 eV). The B_1s_ and N_1s_ peaks were then deconvoluted using the Shirley-type baseline and an iterative least-squared optimization algorithm.

The quality of the ceramic coating was observed via a field emission scanning electron microscope (FESEM, S-4800, Hitachi High-Technologies Corp., Tokyo, Japan), and the elemental composition of the surface was mapped using a SEM-coupled energy dispersive spectrometer (EDS, Oxford Instruments, Abingdon, UK).

#### 2.4.2. Thermal and Electrical Characteristics

The thermal stability of the prepared fibers was examined using a thermogravimetric analyzer (TGA, Shimadzu Corp., Kyoto, Japan). For this purpose, the weight of the samples was collected as a function of temperature change. The sample was heated up to 1000 °C with a heating rate of 10 °C/min under flowing air at 50 cc/min.

The thermal conductivity of PMC samples reinforced with BN–CFs and GF was measured at a steady state, in the vertical direction, to reveal the interfacial effect of ceramic coating on the enhancement of phonon ballistic movement. If achieved, this was expected to subsequently result in a better conduction of energy from the fibrous filler to the composite matrix and vice versa. Thermal conductivity measurement was performed according to the ASTM D5470 method, using the vertical type thermal conductivity measurement system (Hantech Co., Ltd., Gunpo-si, Korea). Before the measurement, the samples were polished until a smooth surface was visually observed. They were then clamped, by about 0.5 kgf, between two thermally conductive polished surfaces (see Figure 2a). Next, a heat flux was imposed to the specimen. The sample was assumed as a thermal barrier with known thickness. The apparatus then measured the thermal impedance (thermal resistance, *R_i_*) of the sample with the assumption of negligible contact resistance between the sample and the conductive surfaces. The resistance of specimens with different thicknesses (*d_i_*) and identical surface areas (*A*) was recorded (see Equation (1)) and plotted against the thickness (see Figure 2b). Thus, the apparent thermal conductivity of the sample could be obtained, as it is equal to the reverse value of the graph slope (see Equation (2)). Several samples were repeatedly measured to ensure the consistency of results.
(1)$$R=\frac{\Delta T}{Aq^{\prime \prime }}=\frac{T_{panel,2}-T_{panel,1}}{Aq^{\prime \prime }}$$
(2)$$k=\frac{1}{\tan\alpha }=\frac{\Delta d}{\Delta RA}=\frac{\Delta d}{\left(R_{sample,2}-R_{sample,1}\right)A}$$
where *k* represents the apparent thermal conductivity.

The electrical resistivity of the composite samples was measured using a conventional four-point probe method (Loresta-GP MCP-T610, Mitsubishi Chemical Analytech Co., Ltd., Yamato, Japan). In order to better clarify the insulating effect of the BN layer on the electrical characteristic of carbon fibers, the electrical resistivity of CF monofilaments was also measured. The error range of electrical resistivity is under 2%.

## 3. Results and Discussion

### 3.1. Surface Characteristics of BN–CFs

The change in the surface morphology of the fibers can be seen in SEM images of as-received and as-prepared CFs shown in Figure 3. All images show the surface morphology of samples along the radial axis and are zoomed in from left to right. They indicate that the fibers were not attached after BN coating, which can be due to a sufficient and non-excessive supply of precursor during the BN synthesis. It could also mean that a good dispersion of coated CFs in PMC can be achievable. Moreover, Figure 4 shows the cross-sectional images of as-received and treated fibers. Crystal-like structures of BN coating were observed in these images. Usually, high crystallinity in solid materials leads to better and higher thermal conduction. SEM images show that the BN coated on CF was not uniformly formed along the axis of the fiber, so it is difficult to confirm the exact coating thickness. It can be roughly estimated that the BN layer has a thickness of about 0.5–0.6 µm.

The change in functional groups present on the fibers was tested via FTIR analysis, and the spectra are shown in Figure 5. The obtained spectrum for as-received fiber confirms the presence of typical O–H stretching (broad), C–H stretching, and C=O stretching absorption peaks at about 3500, 2800, and 1700 cm^−1^, respectively. The O–H stretching bond can be mainly attributed to the hydroxyl (–OH) group formed as a result of the rapid absorption of H_2_O in the air by the potassium bromide, which was used as the matrix during pelleting. The C–H and C=O stretching bonds are the characteristic peaks of the carboxyl (–COOH) group, implying that the CF surface was partially oxidized.

In the case of BN–CF, two peaks corresponding to the formation of B–H stretching and B–N bending bonds were detected at about 2400 and 1380 cm^−1^, respectively [37]. The former belongs to the primary amine (–NH) group, which overlaps with the same wavenumber as the O–H stretching bond on the CF surface. The absorption peak in the middle is the indicator of boron bonded to hydroxyl groups present on the CF surface. Moreover, absorption bands detected at 800 and 1380 cm^−1^ are attributed to the formation of the target coating, i.e., B–N bending vibration [38,39], indicating the synthesis of hexagonal BN (*h*-BN) [40] and turbostratic BN (*t*-BN) [21], respectively [23]. The obtained spectrum for the modified fibers shows the successful synthesis of a chemically bonded BN–CF surface.

The chemical composition of the synthesized coating layer was examined by XPS and EDS analyses. EDS mapping of the fiber surface, shown in Figure 6, reveals the elemental composition of the coating microlayer. It can be seen that, besides the presence of carbon and oxygen atoms, the treated CF surface consists of boron and nitrogen elements, indicating the presence of BN on the CF substrate.

To further confirm the elemental composition of the treated CF surface, XPS spectra (see Figure 7) were obtained and deconvoluted. The elemental composition data are provided in Table 1. The coated CF surface shows extra peaks centered at about 400 and 200 eV, which correspond to nitrogen and boron, respectively [23]. The deconvoluted spectra for asymmetric B_1s_ and N_1s_ bands are shown in Figure 8. Two main peaks were identified at about 190.9 and 398.5 eV corresponding to the formation of B–N and N–B bonds on the treated fiber, respectively [41,42]. As shown in Figure 8a, the presence of boron oxide (B_2_O_3_) species was weakly detected at 192.9 eV [43]. The formation of N–C bonds was confirmed with the respective peak observed at 399.9 eV [42]. Furthermore, the detection of an oxygen peak in BN–CFs can be attributed to the presence of boron oxynitrides (BO_x_N_y_) and the subsequent absorption of oxygen [23]. The XPS results nicely concur with the findings of the FTIR and EDS analyses.

### 3.2. Thermal and Electrical Properties of BN–CFs

The thermal stability of the fibers before and after the modification was examined via TGA. The sample weight was consistently recorded at temperatures ranging from 25 to 1000 °C, and the obtained thermograms are shown in Figure 9. According to these results, the typical high stability and near-zero weight loss of carbon fibers until 640 °C were observed before and after the surface treatment [44]. As reported elsewhere [23], the rise of the temperature above 640 °C shows a slower degrading rate for BN–CFs, as the BN microlayer acted as a protective buffer and was oxidized before the fiber surface. The more gradual oxidation of BN–CFs can be attributed to the formation of a liquid B_2_O_3_ film on the CF surface, acting as an oxygen molecular trap (a diffusion barrier) and a thermal shield [45]. This indicates a higher stability of the coated fiber in harsh environments and a slower decay of the structure when compared with that of a pristine CF. It was assumed that high-temperature treatment of CFs and the formation of BN coating on the fiber surface before TGA analysis formed a thermal shield, consisting of boron oxide and carbon oxide, which delayed the total consumption of the BN-modified CFs when compared with that of the untreated fiber.

Moreover, to evaluate the effect of synthesized BN–CF on heat transfer enhancement in PMCs, samples were examined for their ability to conduct thermal energy carriers, known as phonons. As thermal energy is mainly conducted through lattice vibrations (or phonons) in nonmetal solid materials, phonons are the leading energy carriers in polymer-based composites [46]. Therefore, a proper filler–matrix interface will enhance the motion of crystal lattices within a composite. Thermal conductivity values are shown in Table 2. It was found that the conductivity of the composite reinforced by BN–CF and GF was increased by about 30% when compared with that of the composite reinforced with as-received CF and GF with the same filler content (i.e., 20 phr ea.). This can be ascribed to the lower thermal resistance of the ceramic coating and to the formation of a proper bridge for energy conduction between filler and matrix. Moreover, the surface roughness of CF can be increased after BN modification, resulting in a larger specific surface area. This leads to greater surface energy, which in turn increases the adhesion of fibers to the polymer, facilitating the motion of energy carriers within the composite.

The electrical resistivity values of monofilament fibers are shown in Table 3. As the BN is formed on the surface of the carbon fiber, the difference in resistivity is about 100 times higher than that of the as-received CF monofilament. This further confirms the proper formation of a BN coating layer on the CF surface as well as its effectivity on the electrical resistivity of fibers. Moreover, the results of electrical resistivity for composite samples are shown in Table 2. As seen here, it is also clear that coated fibers become less electrically conductive as BN acts as the electrical buffer layer, preventing the free motion of electrons within the composite.

Overall, the effectivity of the synthesized ceramic coating layer was shown with the results obtained for the thermal and electrical characteristics of the composites. Similar to other ceramic materials, electrons are bound tightly in BN. This leads to an insulating response of the material under electrical charge, which can otherwise be harmful for electronic units. On the other hand, *h*-BN usually, depending on the growth conditions, has a fairly good crystallinity. As explained before, crystallinity leads to a longer phonon mean free path within the material. This means that, unlike electrons, phonons can move more freely in the synthesized ceramic. Moreover, the coating reduces the surface energy gap between the carbon filler and the thermoplastic matrix. This further contributes to the ballistic movement of phonons between the components of the composite system. What is more is the increase in thermal conductivity as well as the reduction in electrical conductivity of composites imply a better dispersion of fillers after the modification of CFs.

## 4. Conclusions

In this study, the synthesis of BN coating on the carbon fiber surface was carried out by means of stepwise solution boriding and high-temperature nitriding of fibers. According to the results obtained from surface analyses, including SEM, FTIR, XPS, and EDS, the fibers were successfully coated without the use of urea in the process. The formation of the expected B–N bond was confirmed by FTIR and XPS spectroscopy methods. The obtained coating was found to increase the final oxidation temperature of CF, as it formed a thermally diffusing layer on the CF surface.

Furthermore, composite samples were fabricated by adding short treated-fibers and graphite powder to a polypropylene matrix. According to the results, BN–CF/GF improved the thermal conductivity by about 316% and 60% compared with those of neat PP and PMC reinforced with as-received CF/GF, respectively. It was found that the synthesized ceramic coating can play a significant role in improving the adhesion of CFs to the polymer, as well as in increasing the thermal stability of fibers in the air. It was also shown that the BN–CF as filler dramatically improved the electrical insulation performance in the composite, i.e., by about 700%.

## Figures and Tables

**Figure 1 polymers-11-02009-f001:**
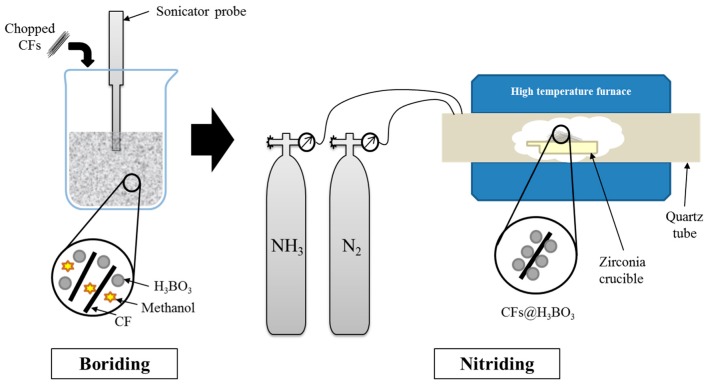
Illustration of the synthesis of the boron nitride (BN) microlayer on the carbon fiber (CF) surface.

**Figure 2 polymers-11-02009-f002:**
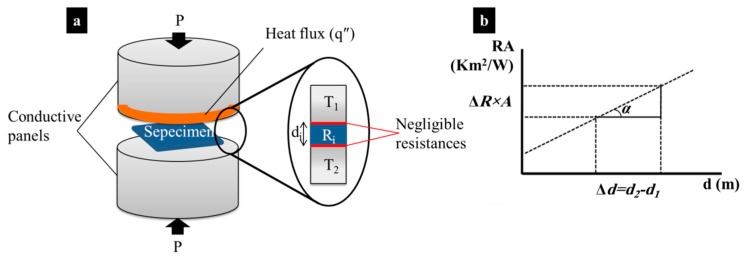
(**a**) Thermal conductivity measuring apparatus and (**b**) the resistance-thickness plot.

**Figure 3 polymers-11-02009-f003:**
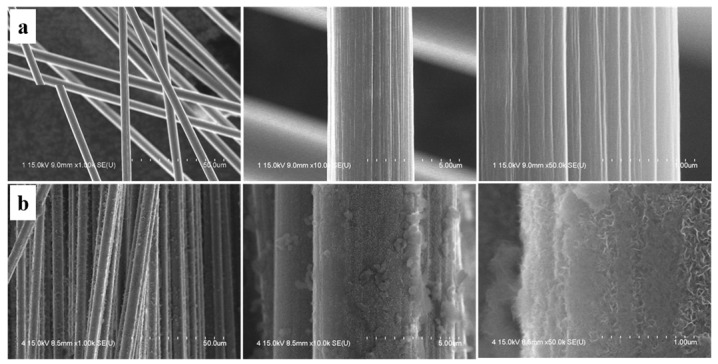
SEM images of the (**a**) as-received CF and (**b**) BN–CF zoomed in from left to right.

**Figure 4 polymers-11-02009-f004:**
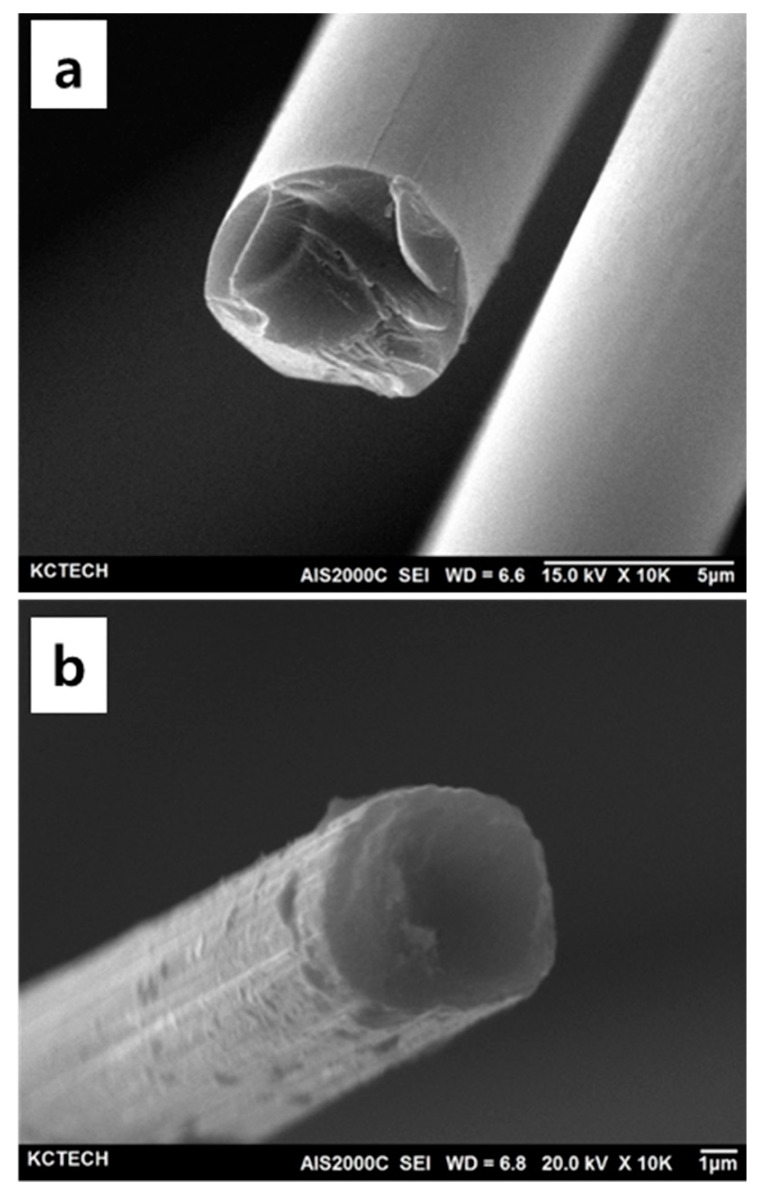
Cross-sectional SEM images of the (**a**) as-received CF and (**b**) BN–CF.

**Figure 5 polymers-11-02009-f005:**
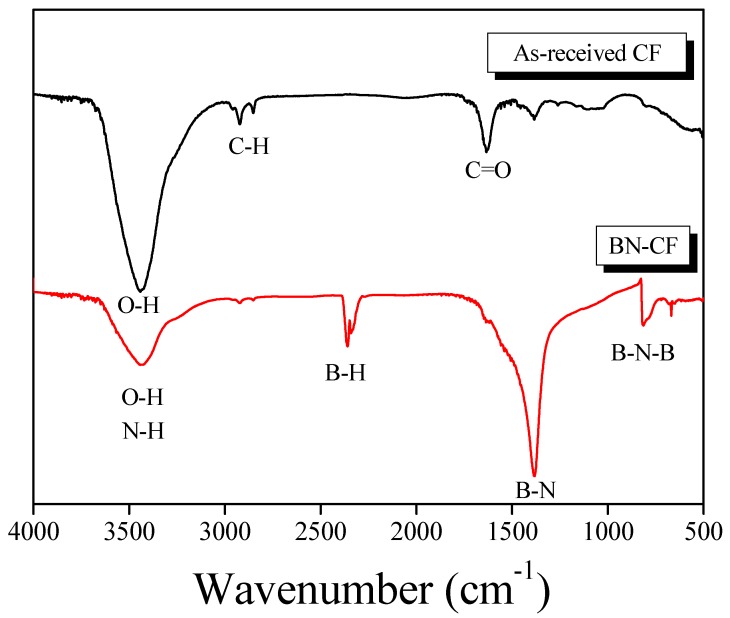
FTIR spectra of the as-received CF and BN–CF.

**Figure 6 polymers-11-02009-f006:**
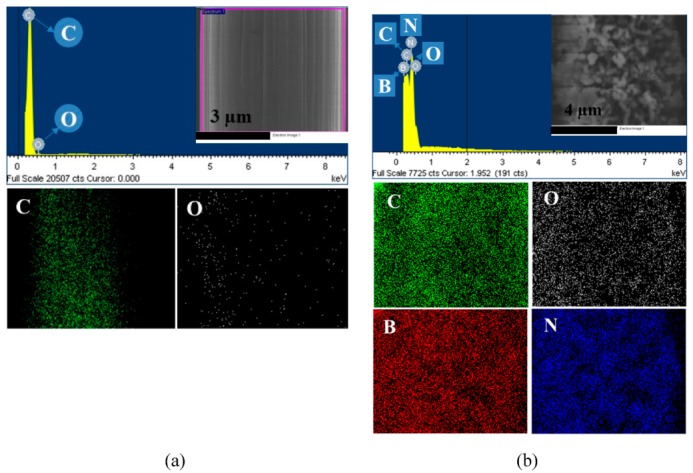
Energy dispersive spectrometer (EDS) mapping images of the (**a**) as-received CF and (**b**) BN–CF with each element.

**Figure 7 polymers-11-02009-f007:**
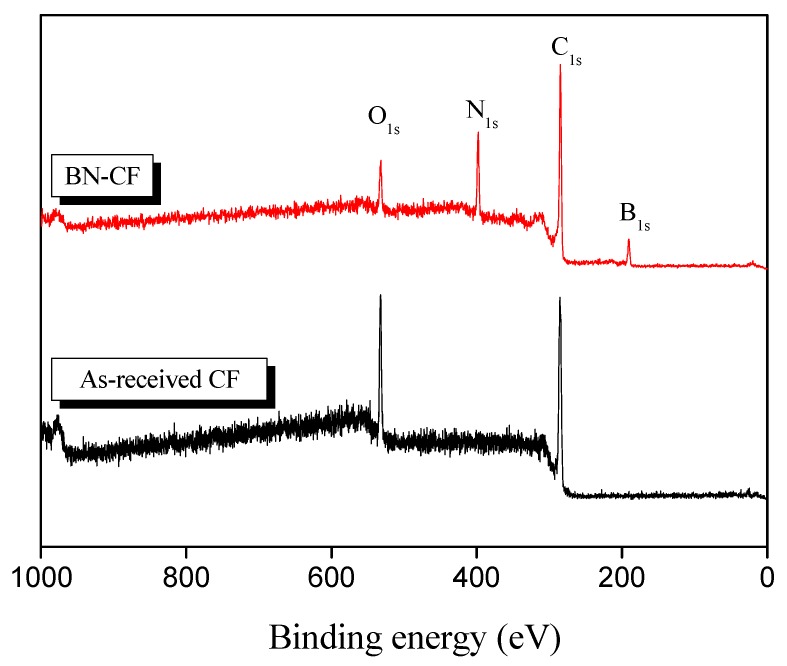
X-ray photoelectron spectroscopy (XPS) spectra of the as-received CF and BN–CF.

**Figure 8 polymers-11-02009-f008:**
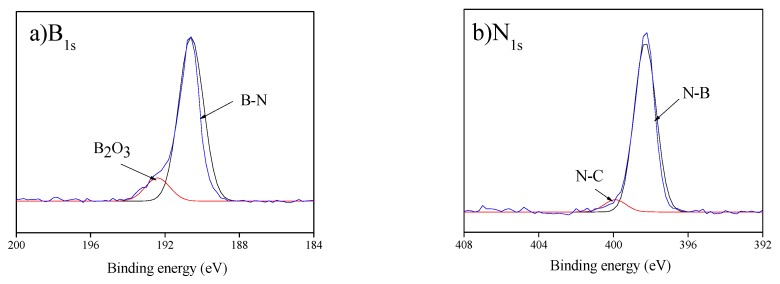
High-resolution (**a**) B_1s_ and (**b**) N_1s_ peaks of the XPS spectrum of BN–CF.

**Figure 9 polymers-11-02009-f009:**
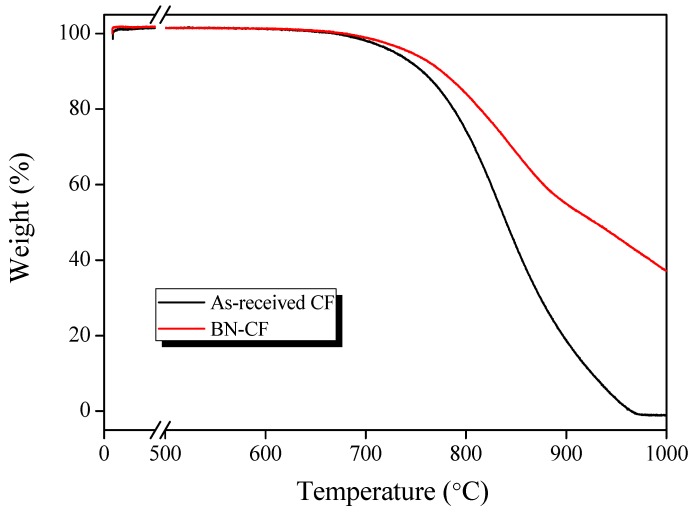
TGA thermograms of the as-received CF and BN–CF.

**Table 1 polymers-11-02009-t001:** Elemental composition of samples.

Sample	Elemental Composition (%)			
	O_1s_	C_1s_	B_1s_	N_1s_
As-received CF	20.55	79.45	-	-
BN–CF	8.30	61.93	14.75	15.02

**Table 2 polymers-11-02009-t002:** Thermal conductivity and electrical resistivity of neat polypropylene (PP) and CF/GF -reinforced PMC samples before and after BN coating.

Sample	Thermal Conductivity (W/mK)	Electrical Resistivity (Ω.cm)
Neat PP	0.18	-
CF20/GF20/PP	0.47	4.72
BN–CF20/GF20/PP	0.75	37.84

**Table 3 polymers-11-02009-t003:** Electrical resistivity of monofilament fibers.

Sample	Electrical Resistivity (Ω.cm)
As-received CF	2.76 × 10^−2^
BN–CF	2.80 × 10^0^

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
