# Peer review of "Preparation of Boron Nitride-Coated Carbon Fibers and Synergistic Improvement of Thermal Conductivity in Their Polypropylene-Matrix Composites"

_polymers, 2019, doi:10.3390/polym11122009_

Round 1

Reviewer 1 Report

The authors reported a new way to preparation BN–CFs with only a gaseous source of nitrogen in a high temperature (1400 °˚C) furnace. This way simplified the synthesis of BN microlayer on CF by removing the biological adverse effects of using urea in the boriding process. In this structure, The BN and GF work together to obtain an electrically and thermally conductive material. The result is great meaning in further synthesizing fillers of composite. However, several problems exist and should be resolved.

The background section needs to cite the newest literature. Many works have reported about the thermal management of the nanocomposites recently, such as: C. Ji, C. Z. Yan, Y. Wang, S. X. Xiong, F. R. Zhou, Y. Y. Li, R. Sun, C. P. Wong, Compos. Part B-Eng. 2019, 163, 363.; C. Z. Yan, T. H. Yu, C. Ji, D. J. Kong, N. Wang, R. Sun, C. P. Wong, Adv. Electron. Mater. 2019, 5, 8.

More details on the measurement of thermal conductivity should be provided such as specific thickness of samples and the press between two conductive panels.

The EDS mapping image (Figure 6) and XPS spectra (Figure 7) of BN-CF shows very higher oxygen content than CF. The authors claim that oxygen is present in the form of B2O3 and BOxNy. However, it is well known that the thermal conductivity and electrical conductivity of the most oxides are very low. Whether the oxygen element affects the performance of composite in this experiment? Is there any way to remove them?

The mechanical properties of the thermal management materials is a very important performance, the authors need to provide relevant data.

“Error! Reference source not found” appears in line 195, 232, 246, which should be corrected.

Reviewer 2 Report

In this work, Badakhsh et al. prepared the boron nitride (BN)–coated carbon fibers (CF) and investigated its effects on thermal conductivity and electrical resistivity of their respective polymer matrix composites. This paper was good written and the results had some important applications. The following should be revised by the authors.

There are lot of grammatical and cited errors in the manuscript.

In the Introduction section, more references published in recent years should be cited. In addition, recent literature should be cited by giving explanation of comparison of your work.

Author prepared PP composite samples with only one content (CF20 and BN-CF20), only one data points are not suitable for comparison.

In the abstract, the authors mentioned that “The composite characterization indicates approximately 30% improvement in through-plane thermal conductivity and about 700% increase in electrical resistivity of sample containing the 20 phr of BN–CF”. Please indicate, also in the text?

Reviewer 3 Report

In this article, authors have reported some important results on PP based composites. Following corrections are needed in the article:

-Fabrication of PMC Samples should be explained in more detail

-Specific reasons for the selection of heating rate 20 °C/min should be given.

-What is the conc. of boric acid used?

-Figure 3 should be reproduced as images are not very clear

-Authors should consider citing relevant articles on PP and boron-based nanomaterials such as ACS applied materials & interfaces 6 (12), 9349-9356 (2014); Polymer Chemistry 3 (4), 962-969  (2012); Vacuum 146, 641-648 (2017)

Round 2

Reviewer 2 Report

There are still problems, such as

Please explain the relationship between filler graphite flakes (GFs) and natural graphite (NG) powder in "2.1. Materials". Please give the specific model of high-temperature nitriding instrument and internal mixer. The author needs to supplement the SEM characterization and corresponding analysis of CF/GF/PP and BN-CF /GF/PP. Please explain why the BN coated on the carbon fiber is not uniformly coated, and how to estimated that the BN layer has an approximate thickness of 0.5-0.6 µm. Please compare the effect of adding different components of BN. Please explain the mechanism of utilizing BN–CF incorporated with GF as the reinforcing fillers, and compare the difference between BN–CF with GF and without GF. The author can consider making more graphs when doing the characterization of thermal conductivity and Electrical resistivity, so that the data can be more reliable. In addition, the experiment and analysis of thermal and electrical properties of BN-CFs is too brief. Some properties can be added into this article like permittivity. The ordinate content in Figure 9 is wrong. The ordinate should be weight rather than weight loss. The experimental conditions of thermogravimetric analysis should be given more detailed. Please explain why the two-dimensional structure of GFs can reduce the prepenetration value and why the amount of graphene in the experiment is 20phr.
